# Moisture control of tropical cyclones in high-resolution simulations of paleoclimate and future climate

Pavan Harika Raavi[1,2], Jung-Eun Chu [1,3] ✉, Axel Timmermann[1,4], Sun-Seon Lee [1,4] & Kevin J. E. Walsh[5]

The intensity of tropical cyclones (TCs) is expected to increase in response to greenhouse warming. However, how future climate change will affect TC frequencies and tracks is still under debate. Here, to further elucidate the underlying sensitivities and mechanisms, we study TCs response to different past and future climate forcings. Using a high-resolution TC-resolving global Earth system model with 1/4° atmosphere and 1/10° ocean resolution, we conducted a series of paleo-time-slice and future greenhouse warming simulations targeting the last interglacial (Marine Isotope Stage (MIS) 5e, 125 ka), glacial sub-stage MIS5d (115 ka), present-day (PD), and $CO_2$ doubling ($2 \times CO_2$) conditions. Our analysis reveals that precessional forcing created an interhemispheric difference in simulated TC densities, whereas future $CO_2$ forcing impacts both hemispheres in the same direction. In both cases, we find that TC genesis frequency, density, and intensity are primarily controlled by changes in tropospheric thermal and moisture structure, exhibiting a clear reduction in TC genesis density in warmer hemispheres.

Landfalling tropical cyclones (TCs) can cause severe damage and impact human livelihoods and lives. Understanding what controls their frequency, tracks, and intensities is therefore of utmost importance. There is an emerging consensus between numerical models and theory suggesting that greenhouse warming will likely increase the intensity of TCs in the future[1–6]. However, there is still a large uncertainty in model-based projections of TC frequencies. Part of this uncertainty can be attributed to sensitivities with respect to external forcings, climate model physics, and TC tracking algorithms[4,7–9]. Due to a lack of theoretical understanding, it is unclear whether and where future TC frequencies will increase or decrease. In general, paleoclimate studies may provide an additional testbed for TC activity changes in future projections.

To gain a deeper understanding of how TCs respond to warm background climate states, several studies have examined synoptic and mesoscale atmospheric variability during the past warm climate episodes such as the Last interglacial period (LIG, ~125 thousand years before present, ka, global mean annual surface temperatures 0.8–0.13 °C warmer than pre-industrial, and $CO_2$ ~ 280 ppm)[10], the Eocene (56,000–33,900 ka, global mean surface temperatures 10.4–15.6 °C warmer than later 20th century and $CO_2$ ~ 1400 ppm), and the Pliocene (5300–2600 ka, global mean surface temperatures 1.8–3.6 °C warmer than pre-industrial and $CO_2$ ~ 400 ppm)[11]. The LIG period was the latest time when global mean surface temperatures exceeded pre-industrial level; however, the $CO_2$ levels were lower than other hothouse periods such as the Eocene and Pliocene, and the elevated temperatures during the LIG were likely caused by strong seasonal variations in the incoming solar radiation and corresponding climate rectifications[12].

Recent modeling studies[11,13,14] have documented that past hothouse boundary conditions allowed TCs to form at higher latitudes, as

[1]Center for Climate Physics, Institute for Basic Science (IBS), Busan 46241, Republic of Korea. [2]Centre for Climate Research Singapore (CCRS), Singapore, Singapore. [3]Low-Carbon and Climate Impact Research Centre, School of Energy and Environment, City University of Hong Kong, Hong Kong, China. [4]Pusan National University, Busan 46241, Republic of Korea. [5]School of Geography, Earth and Atmospheric Sciences, University of Melbourne, Parkville, Australia. ✉e-mail: jungeun.chu@cityu.edu.hk

projected for future warmer conditions. However, whether any of these paleo situations can be regarded as a "qualitative analog" for the future is still an open question. This is because those TCs from paleoclimate simulations are not only affected by greenhouse gas forcing but also by orbital forcing, which incorporates seasonal variations in incoming solar radiation because of changes in Earth's orbit and axis. Previous results on the effect of orbital forcing reveal that when the solar radiation is stronger in the Northern Hemisphere (NH), the NH becomes less favorable for TC activity, while the Southern Hemisphere (SH) becomes more favorable[15,16]. Nevertheless, most of the studies are based on the idea that large-scale environments such as TC genesis potential indices (GPIs) can represent changes in TC activity. To obtain a detailed perspective on the relationship between large-scale climatic drivers and TC statistics, it is important to explicitly resolve TCs and their key mesoscale features in a climate model. This requires a relatively high atmospheric model resolution of at least 50 km[17] and it is also desirable to capture the coupling between atmosphere and ocean to account for cold wake effects and changes in ocean heat content, which may further influence the character and the statistics of TCs[18]. To the best of our knowledge, there are no systematic paleo-time-slice simulations that explicitly resolve TCs using a fully coupled high-resolution Earth system model. In other words, TC statistics such as frequency and intensity under different background conditions have not been explored in detail yet. Therefore, our study aims to better comprehend TC statistics for past and future warmer climates by using high-resolution Earth system models that explicitly simulate TCs.

Apart from the question of "analog", it is also important to elucidate how TCs are formed under different boundary conditions[15,16] to identify the underlying physical mechanisms and controlling factors. The LIG, also referred to as Marine Isotope Stage (MIS) 5e (-125 ka), and MIS5d (the glacial sub-stage, -115 ka), with large seasonal changes in solar radiation, serve as excellent test grounds. For consistency, we will refer to the LIG as MIS5e hereafter. These two periods, MIS5e and MIS5d, represent extreme orbital conditions with low and high precession indexes, respectively, and large eccentricity. A low (high) precession corresponds to NH summer perihelion (aphelion) conditions and associated intensified (weakened) NH summer solar radiation. This effect is visible only for high values of Earth's orbit eccentricity, and the interval from 125 to 115 ka represents one of these periods. In other words, contrary to the present-day (PD) orbital configuration, the MIS5e is characterized by a warmer NH summer and a colder SH summer. The MIS5d is characterized by a warmer SH summer and a colder NH summer. Understanding the TC response to an altered seasonal cycle amplitude may shed further light onto the roles of interhemispheric forcings and seasonal changes in environmental factors such as sea surface temperature (SST), humidity, and vertical wind shear in TC genesis.

Here we set out to study the role of extreme orbital conditions and greenhouse gas forcing on TC frequencies and intensities using the high-resolution version[19] of the Community Earth System Model (CESM) 1.2.2 with 0.25° atmosphere and 0.1° ocean resolutions. To this end, we conducted two century-long time-slice simulations for MIS5e (-125 ka) and MIS5d (-115 ka) conditions (Fig. 1A) ("Methods"). The orbital forcing simulations are then compared with the experiments for PD climate and $CO_2$ doubling conditions ($2\times CO_2$)[6,20,21]. Compared to the previous study[6], which used the last 20 years of the simulation, this study detects TCs using the last 60 years of each simulation. We also employ two fundamentally different TC tracking algorithms (i.e., traditional vortex-based atmospheric conditions and phenomenon-based large-scale conditions; "Methods" text) to evaluate the sensitivity of the TC tracking schemes. To identify the linkage between large-scale and synoptic conditions, the results are then compared with the changes in the genesis potential index (GPI) and each of its controlling factors, also referred to as GPI component analysis. This

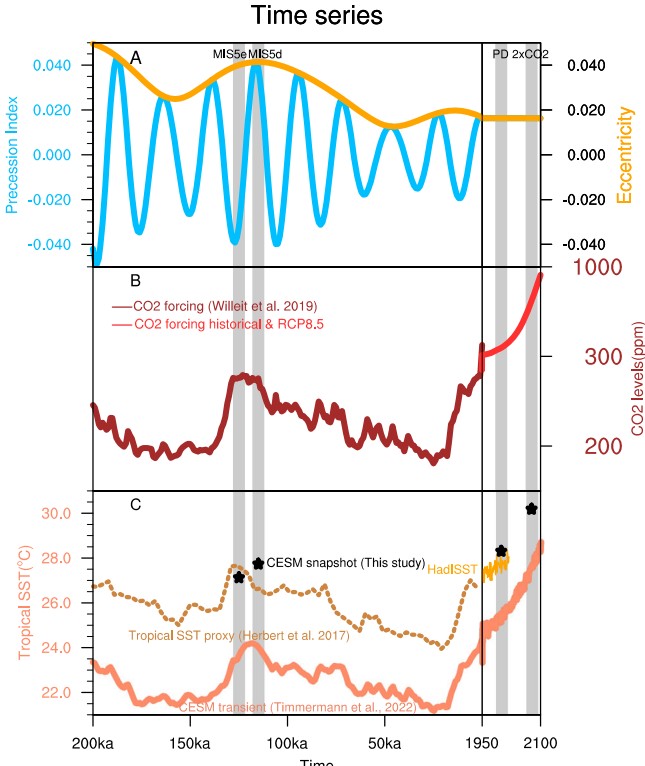

**Fig. 1 | Time series of the precession index, eccentricity, atmospheric $CO_2$ concentration levels (ppm), and sea surface temperatures (SSTs). A** Time series of precession index (sky-blue line), $e\sin(\omega_s)$, where $e$ is the Earth's orbital eccentricity (light-orange) and $\omega_s$ quantifies how close the Sun is to the Earth at midsummer from 200 ka (paleoclimate) to 2100 (future climate), **B** time series of $CO_2$ concentration levels from a low-resolution intermediate complexity model simulation (dark red)[54] and historical and Representative Concentration Pathway (RCP) 8.5 scenario (light red), and **C** tropical SSTs (°C) simulated by the low-resolution Community Earth System Model (CESM)[23] (averaged between 5°S–5°N; solid orange) along with proxy SST reconstructions from selected sites (dashed orange line)[22] and modern SSTs from the HadISST dataset (light-orange line after 1950). Stars indicate the tropical SST estimated within 5°S–5°N using the current high-resolution CESM during the times of MIS5e, MIS5d, and PD. The timing for time-slice simulations is indicated by gray bars.

study aims to determine whether there is a common link between large-scale environments and changes in TC characteristics from the past to the future. It also eventually seeks to improve the robustness of our physical understanding of the dominant drivers of TC frequency and intensity changes under various background conditions.

## Results

### Climate response to orbital and greenhouse gas forcings

MIS5e and MIS5d correspond approximately to the minimum and maximum phase of the precession index (Fig. 1A) during the past 130 ka. Atmospheric $CO_2$ concentrations of ~275 ppm in both periods (Fig. 1B), as well as the corresponding mean tropical temperatures, were similar to pre-industrial levels (Table 1, Fig. 1C), as shown by a reconstruction of tropical SST[22] and a transient climate model simulation[23] conducted with a lower-resolution model version of CESM1.2.2. Therefore, the comparison between the corresponding CESM1.2.2 paleo simulations reveals mostly the impact of precessional variability and the associated effect of seasonal and hemispheric insolation asymmetry on the climate system.

Analyses of averaged tropical climate variables relevant for TC dynamics over the oceans in these simulations reveal the following: atmospheric responses to incoming solar radiation in the orbital

**Table 1 | Model experimental details of atmospheric gases, orbital year, and surface data**

| Model parameters | Orbital forcing experiments | | Greenhouse gases forcing experiments | |
|---|---|---|---|---|
| | Marine Isotope Stage 5e (MIS5e) | Marine Isotope Stage 5d (MIS5d) | Present-day (PD) | Future (2×CO₂) |
| Orbital year | 125 ka | 115 ka | 1990 | 1990 |
| $CO_2$ concentration (ppm) | 276.0 | 275.6 | 367.0 | 734.0 |
| Other greenhouse gases (ppm) | $CH_4$ = 640.4e-9 <br> CFC-11 = 0.0 <br> CFC-12 = 0.0 <br> $N_2O$ = 263.1e-9 | $CH_4$ = 472.0e-9 <br> CFC-11 = 0.0 <br> CFC-12 = 0.0 <br> $N_2O$ = 251.3e-9 | $CH_4$ = 1760.0e-9 <br> CFC-11 = 653.5e-12 <br> CFC-12 = 535.0e-12 <br> $N_2O$ = 316.0e-9 | $CH_4$ = 1760.0e-9 <br> CFC-11 = 653.5e-12 <br> CFC-12 = 535.0e-12 <br> $N_2O$ = 316.0e-9 |
| Aerosol forcing | Pre-Industrial (PI) condition | | PD condition | |
| Surface data | PI condition | | PD condition | |
| Urban effect | No | | Yes | |
| Crop | No | | Yes | |

forcing simulations during the NH summer and SH summer are out of phase (Fig. 2A). Due to the changing seasonal insolation by precession, there is increased solar radiation in the NH summer (July to September) and decreased radiation in the SH summer (January to March) during MIS5e compared to MIS5d (blue line in Fig. 2A). This anomalous distribution of solar insolation generates a warmer NH summer and autumn (as in the 2xCO₂ forcing simulation at 200 hPa) and a colder SH summer and autumn (blue solid and dashed lines Fig. 2B). We observe increased temperatures with a lag of a few months both at lower and upper levels with a greater magnitude in the upper levels (Fig. 2B). The forced specific humidity variations are proportional to the temperature changes with higher values during the NH summer and lower values during the SH summer months in the MIS5e in comparison with MIS5d (Fig. 2C). As warm air can hold more moisture, the region with increased temperatures should, in general, correspond to reduced relative humidity (RH). Interestingly, the diagnosed changes in RH show a decrease during the NH summer and an increase during the SH summer in the MIS5e compared with the MIS5d simulation, however, the peak season is not exactly in phase with temperature (Fig. 2D). Especially, there is no apparent trend in decreasing RH in the NH summer due to regional variations in the moisture content associated with the regional-scale phenomenon (Supplementary Fig. S1).

In contrast to the orbital forcing simulations, which have clear seasonal (i.e., hemispheric) differences in the temperature and moisture responses, the 2×CO₂ forcing simulation exhibits annually and hemispherically symmetric positive temperature anomalies relative to the PD climate conditions (red solid and dashed lines in Fig. 2B). The increased CO₂ concentration in the atmosphere increases tropospheric temperatures with significantly higher values in the upper levels compared to the lower levels, due to the moist adiabat response (Fig. 2B)[13,24-26]. The increased global air temperatures lead to year-round increased values of specific humidity and reduced RH in the lower troposphere across the tropical ocean basins (red line in Fig. 2C, D). Although there are variations in each month, it is interesting to observe that the NH summer climatic response to orbital forcing (i.e., MIS5e minus MIS5d) is qualitatively equivalent to that of the NH summer greenhouse warming response with higher temperatures. On the other hand, the SH summer response to orbital forcing is opposite to that of the CO₂ doubling forcing (Fig. 2B).

These differences in temperatures and moisture content of the atmosphere lead to changes in the stability and the moist-entropy deficit parameters (Supplementary Fig. S2). It has previously been demonstrated that large-scale increases in atmospheric stability led to a reduction in TC frequency[27-29]. In addition, increased values of the moist-entropy deficit provide unfavorable conditions for TC formation, which reduces the extent of the sustained atmospheric convection during the genesis stage of a storm[30,31]. The positive stability and moist-entropy deficit anomalies compared to the PD are most pronounced for the 2×CO₂ simulation and they affect both hemispheres in

their respective summer season. In contrast, in the MIS5e simulation, we observe higher stability and moist energy deficit during the NH summer as compared to the MIS5d simulation (Supplementary Fig. S2A, C); but weaker values during the SH summer (Supplementary Fig. S2B, D). Therefore, the environmental response to greenhouse gas forcing allows us to simply expect reduced TC frequencies in both hemispheres. In contrast, a hemispheric asymmetric response to orbital forcing is expected with reduced TC frequency in the NH summer and enhanced TC frequency in the SH summer. We will further test these hypotheses using the GPI analysis compared with two TC tracking schemes and explore whether the MIS5e NH summer can be regarded as a qualitative analog to greenhouse warming.

**Large-scale TC genesis environments**

To understand the impact of orbital forcing and greenhouse gas forcing on TC genesis environments, we calculate the GPI (see Supplementary Fig. S3 and "Methods") for different climate conditions from paleoclimate to future climate[16,31-36]. We also evaluate the relative importance of each GPI component in relation to the corresponding change in the large-scale environment for TC genesis (i.e., GPI component analysis, Fig. 3 and Supplementary Fig. S4). As peak TC activity seasons vary by basins, we calculated GPI changes in the peak TC seasons for the respective basins (July to September (JAS) for the North Atlantic (NA), Eastern North Pacific (ENP), and Western North Pacific (WNP); October to December (OND) for the North Indian (NI); and January to March (JFM) for the Southern Indian (SI) and Southern Pacific (SP) basins), and combined them in one map (Fig. 3). In response to the low precession forcing, the total GPI is reduced in key regions of TC genesis such as the WNP, the northwestern NA, and equatorial side of the ENP, while is enhanced in the NI and SI, subtropical ENP, Caribbean Sea, and some areas in the SP (Fig. 3A). In the paleoclimate simulations, the thermodynamical conditions (i.e., moist-entropy deficit and maximum potential intensity) can explain most of the GPI changes except for the Caribbean Sea and north of WNP region (20°N–40°N) where dynamical conditions (absolute vorticity and vertical wind shear) largely contribute to the changes in GPI. Meanwhile, in the greenhouse gas forcing experiment, the total GPI is reduced over almost all the key TC genesis regions (Fig. 3F), and the moist-entropy deficit explains most of the changes in TC frequency (Fig. 3I). The percent contribution of each GPI term to the basin-averaged total GPI change is provided in Table 2. For both the orbital forcing and 2×CO₂ simulations, the diagnosed changes in GPI are in first-order consistent with the changes in the thermodynamic variables (Table 2). Other GPI factors either play an important role on the regional scale or a secondary role in explaining the large-scale shifts in TC genesis potential GPI. In contrast, a recent paleo-study focusing on the Paleocene–Eocene thermal maximum (PETM) epoch (characterized by increased CO₂ conditions and land-sea distribution) found that changes in vertical wind shear can explain the changes in the TC

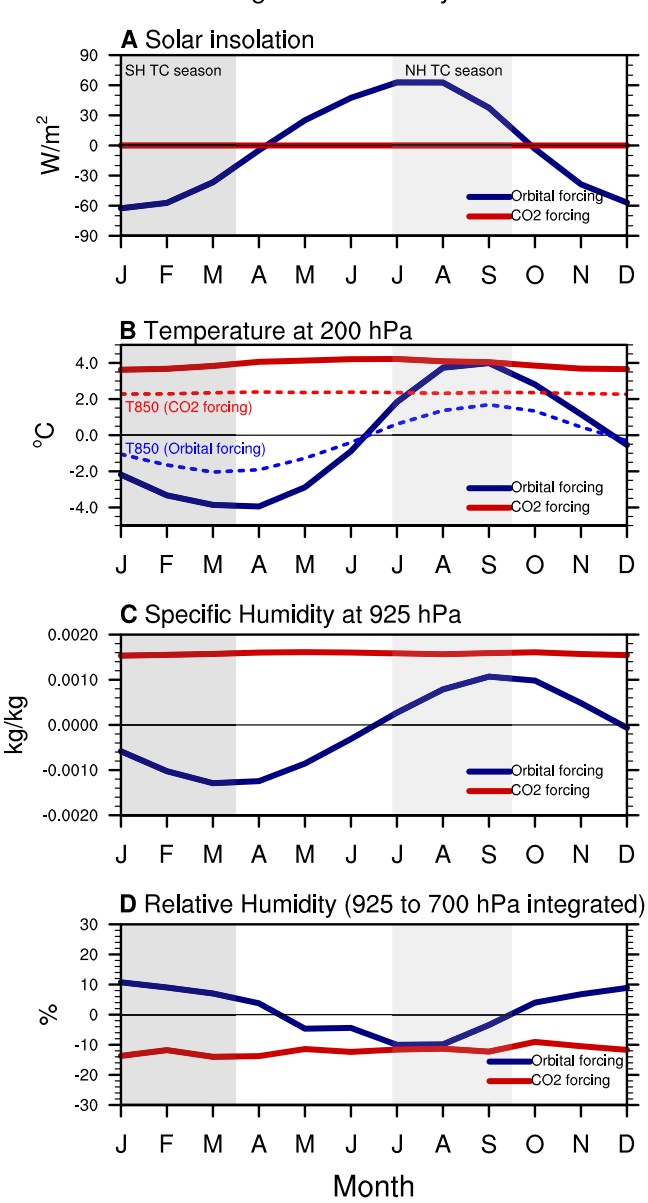

**Fig. 2 | Changes in the annual cycle due to orbital forcing (MIS5e minus MIS5d) and CO₂ forcing (2×CO₂ minus PD) in the tropics. A** Incoming solar radiation, **B** temperature at 200 hPa (solid line) and 850 hPa (dashed line), **C** specific humidity at 925 hPa, and **D** vertically integrated low-level relative humidity from 925 to 700 hPa. The blue line indicates the annual cycle difference between MIS5e and MIS5d, and the red line indicates that between 2×CO₂ and PD. The figures are obtained by averaging the respective quantities over ocean points between 30°N to 30°S. Light gray shading indicates the Northern Hemisphere (NH) tropical cyclone (TC) season (July to September, JAS), and dark gray shading indicates the Southern Hemisphere (SH) TC season (January to March, JFM).

genesis under increased greenhouse gas conditions[13]. In our orbital forcing simulations, the vertical wind shear is also not negligible. Their overall contribution to the entire basin, however, is less significant than thermodynamic variables because it is limited to the equatorial zone (Eq to 10°N) and out-of-phase in the equatorial WNP. Furthermore, the contribution of vertical wind shear in response to greenhouse gas forcing is not statistically significant except for ENP and NI.

To further test the robustness of our results, we also calculated the conventional GPI equation[37] (see "Methods"), which differs from Korty's approach in two terms: (1) RH at 700 hPa instead of moist-entropy deficit and (2) vertical velocity at 500 hPa. The conventional

GPI component analysis also shows that moisture-related variables, i.e., 700 hPa RH, can capture most of the changes in the GPI (Supplementary Fig. S4), thereby supporting our results from the other GPI equation.

Reduced values of RH and increased stability of the atmosphere due to increased upper-level temperatures act as unfavorable conditions for TC formation. The RH is an indicator for water vapor given the maximum amount of water the air can hold at a temperature. Therefore, it is proportional to water vapor in the atmosphere and approximately inversely proportional to atmospheric temperature. In differences between the two paleoclimate simulations, we observe a decrease in global mean RH during the NH warm season and an increase during its cold season, indicating that temperature plays a larger role in determining the sign of RH (Supplementary Fig. S5A–C). In the greenhouse warming experiment, the monthly global mean RH decreases with increasing temperatures (Supplementary Fig. S5D–F). The RH in the orbital forcing simulations shows regional differences (Supplementary Fig. S1). The changes in the temperature and moisture across the NA, WNP, and all the SH ocean basins agree with the global changes, but the ENP and NI basins have higher RH during the NH summer, irrespective of the higher air temperatures. We notice stronger low-level monsoon westerlies in the MIS5e simulation (Supplementary Fig. S6D), which can be partly related to the enhanced land-sea thermal contrast[38]. This circulation is associated with increased moisture transport into the Arabian Sea region as compared to the Bay of Bengal region (Supplementary Fig. S6B). In addition, across the ENP and NI basins, we observe a northward shift of the intertropical convergence zone (ITCZ) position in the MIS5e simulation leading to favorable conditions for TC formation (Supplementary Fig. S7B, E).

### Changes in detected TC genesis
The aforementioned changes in the large-scale conditions of different model simulations (i.e., orbital and CO₂ forcing) can lead to changes in explicitly detected TCs as the storm tracking schemes use thresholds dependent on background large-scale environmental conditions. One of the advantages of high-resolution, fully coupled simulations presented here is that they explicitly resolve the structure of TCs and relevant air-sea interactions[6]. As illustrated here for a category-4 storm (Saffir-Simpson) in the MIS5e simulation (Supplementary Fig. S8), TCs have a pronounced eye in the center and an SST cold wake effect oftentimes larger than 3 °C. It is also noted that the current version of the CESM under PD conditions can capture the observed TC genesis climatology with certain regional variations while underestimating TCs in the NA and WNP basins[6]. To study the effect of paleoclimate and future forcings on TC tracks, we detected TCs using the last 60 years of each simulation (Supplementary Fig. S9, "Methods").

Changes in TC genesis density during MIS5e show an increased density in the SH summer across southwestern SI and SP ocean basins compared to the MIS5d (Fig. 4A), mostly due to favorable TC forming conditions. Although not represented by the GPI, the tropical eastern Indian Ocean near Sumatra Island has a lower genesis density (Fig. 4A). In addition, during the MIS5e period, the genesis density and track density differences (Fig. 4A, C) show a reduced number of detections in NA and WNP regions whereas the NI (Arabian Sea) and ENP basins show higher frequencies. The global and annual mean number of TCs in MIS5e rises by 5% in comparison to that in MIS5d as a result of a 3% reduction in the NH and an 18% increase in the SH (Supplementary Fig. S9E). We found no clear shift in the annual cycle (Supplementary Fig. S10).

The simulated changes in global and hemisphere mean TC genesis density (Fig. 4) agree with other paleoclimate simulations conducted with a coarser-resolution model that employs GPIs to study the variations of TC formations during the mid-Holocene period[15,16]. However, these studies did not identify the inter-basin differences in genesis

## MIS5e-MIS5d

## 2xCO2-PD

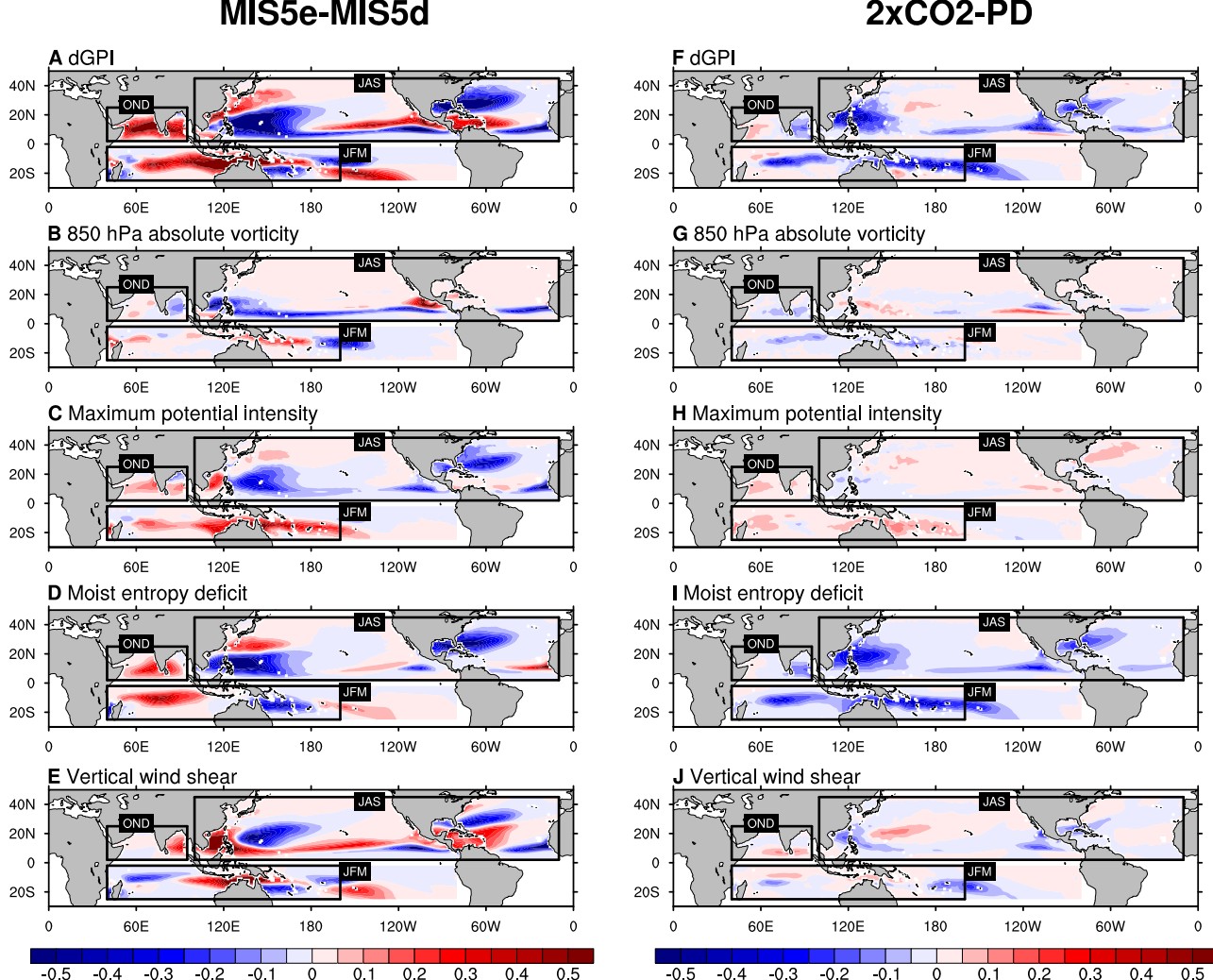

**Fig. 3 | Changes in Genesis potential index (GPI) components from the GPI variational component analysis.** Changes in GPI components from **A–E** orbital forcing simulations and **F–J** increased $CO_2$ concentration simulations, using the high-resolution CESM 1.2.2. We used simulated large-scale variables in the paleoclimate and future climates averaged for a period of 60 years from each simulation during the TC peak seasons across different ocean basins (i.e., JAS in North Pacific and North Atlantic basins; JFM in SH basins; October to December (OND) in North Indian Ocean).

**Table 2 | Percentage contribution of each genesis potential index (GPI) terms[15] ("Methods") to total basin-averaged changes in GPI in response to orbital forcing (MIS5e-MIS5d) and greenhouse gas forcing (2×CO2-PD)**

| Relative contribution (%) | MIS5e-MIS5d | | | | 2×CO$_2$-PD | | | |
|---|---|---|---|---|---|---|---|---|
| | Term1 | Term2 | Term3 | Term4 | Term1 | Term2 | Term3 | Term4 |
| NA (JAS) [290°E-350°E, EQ-35°N] | 15.6 | 42.8* | 24.3* | 17.3* | 11.3 | −18.6 | 88.7* | 18.6 |
| ENP (JAS) [240°E-260°E, EQ-20°N] | −9.2* | 42.9* | 25.7 | 40.5* | 6.7 | −12.6* | 60.0* | 45.9* |
| WNP (JAS) [100°E-180°E, EQ-40°N] | 28.2 | 29.2* | 53.3* | −10.7* | −5.5 | 2.9 | 89.6* | 13.0 |
| NI (OND) [45°E-100°E, EQ-30°N] | −7.7 | 26.3* | 48.2* | 33.3* | 184.1 | −214.8* | 302.2* | −171.6* |
| SI (JFM) [30°E-135°E, 30°S-EQ] | 14.3 | 38.7* | 37.7* | 9.3* | 40.1 | −71.5* | 151.5* | −20.1 |
| SP (JFM) [135°E-290°E, 30°S-EQ] | −39.5 | 202.2 | −104.3 | 41.6* | 0.4 | −16.1* | 83.7 | 32.0 |

The percentage contribution is calculated by dividing the relative role of each GPI term (each term on the right-hand side of the GPI component analysis) by the total changes in GPI. Stars indicate the 95% significance level based on the Student's *t*-test.

frequency and track density. Although there is a regional discrepancy between TC density and GPI in the SH summer during the MIS5e, the general changes in TC genesis density during the NH summer are consistent with the changes in the GPI.

The future climate (i.e., greenhouse gas forcing) simulation shows reduced global annual TC genesis frequencies in both hemispheres compared to the PD (Fig. 4B and Supplementary Fig. S9)[6]. The track densities indicate decreased densities in the tropics with a slightly increased track density in higher latitudes of the NH basins and in some locations of the South Pacific basin (Fig. 4D). In contrast to the MIS5e climate that exhibits a clear hemispheric asymmetry in the genesis frequency, the future climate simulation shows an overall decrease in the mean annual TC frequency across all the ocean basins in both hemispheres. Previous studies demonstrated that changes in

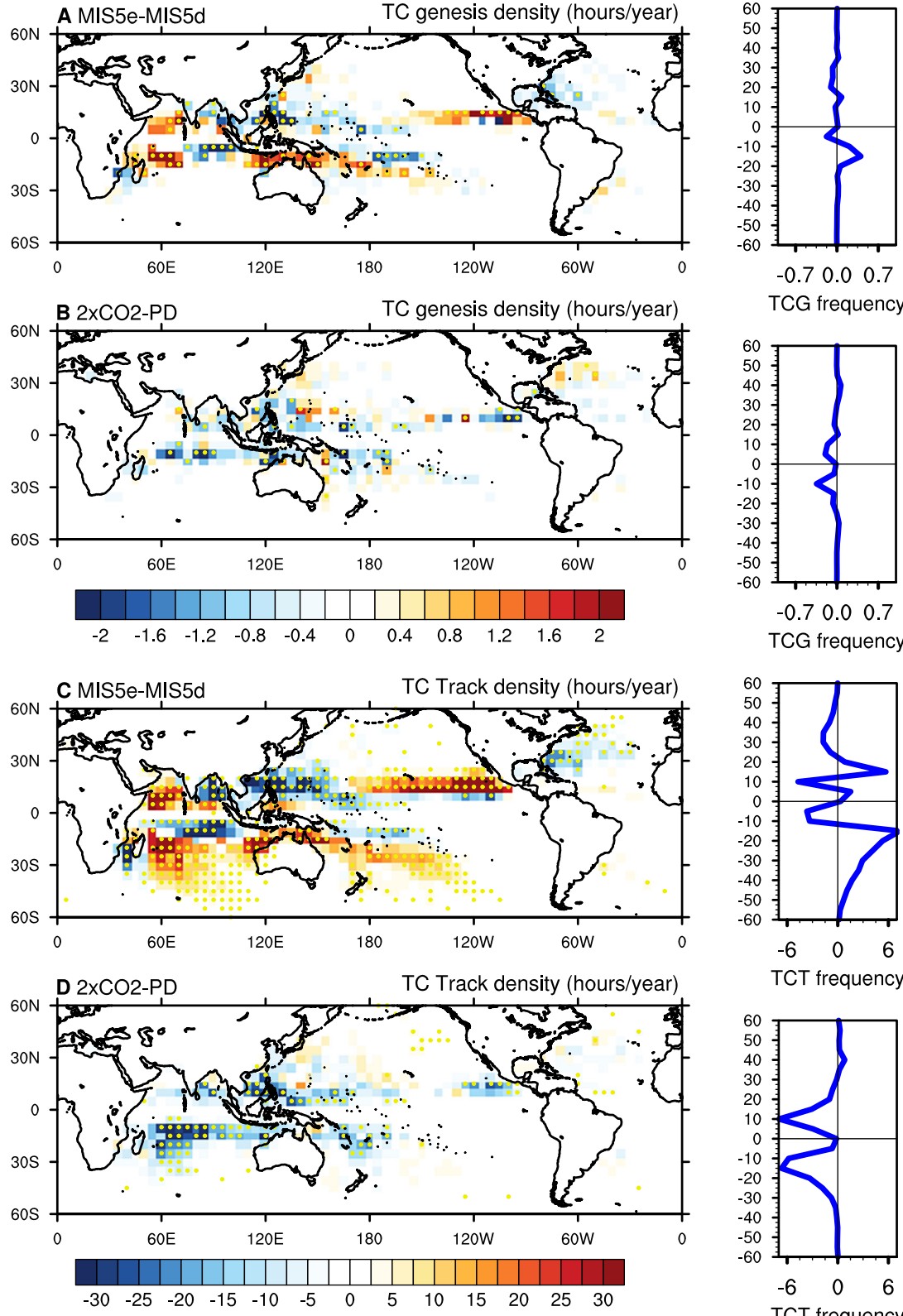

**Fig. 4 | Spatial (shadings) and latitudinal (blue solid line) differences in TC genesis and track densities estimated with a 5° × 5° grid box per year.** **A**, **C** Paleoclimate simulations (MIS5e minus MIS5d), and **B**, **D** present-day and future climate simulations (2×CO₂ minus PD) for **A**, **B** genesis density and **C**, **D** track density. Yellow dots indicate the statistically significant (95%) values measured using a two-sample *t*-test. The latitudinal distribution of zonally averaged TC genesis (TCG) and TC track (TCT) frequency changes are displayed in blue line plots on the right panel.

the Hadley circulation (i.e., weakening of rising branches of Hadley cells in both the hemispheres) along with unfavorable environmental conditions could suppress TC formation in both hemispheres in response to $CO_2$ forcing[4,6,11,39]. In addition, the changes in TC genesis locations and tracks are insensitive to the use of traditional and phenomenon-based tracking schemes both in the paleoclimate and future climate, thereby increasing the confidence of the simulated TC response (Supplementary Fig. S11 and "Methods").

It is interesting to note that the changes in GPI are qualitatively similar to the detected changes in TC genesis that are explicitly simulated by the model using different tracking schemes in most of the ocean basins. Previous studies on the GPI analysis showed that there is some disagreement between the model-simulated TCs and GPI-estimated TC genesis frequency in future warmer climates[40]. However, in the current study, the considerable agreement of these GPI results in most of the ocean basins using two GPI indices, including the paleoclimate, present, and future climates with the explicitly simulated TCs by the high-resolution model is encouraging. Additionally, unlike the response to a future warmer climate, the TC genesis density is not strongly reduced in the NH summer during the paleo-warmer period (MIS5e).

We further compare the locations of TC genesis detected using TC tracking schemes to seasonal changes in the thermodynamical and dynamical conditions that influence TC formation. The seasonal variations in the 700 hPa RH closely align with the TC genesis frequency changes in most of the ocean basins showing that moisture may be a key controlling factor for the orbitally induced changes between MIS5e and MIS5d (Supplementary Fig. S6B). Similarly, another thermodynamical variable, maximum potential intensity, can capture the detected TC genesis frequency in most of the basins except ENP (Supplementary Fig. S6E). Moving on to changes in the dynamical variables, the development of the localized cyclonic vortex depends on large-scale absolute vorticity in the lower atmosphere. The seasonal fluctuations in the vorticity between the MIS5e and MIS5d are consistent with variations in the TC frequency, especially in SH basins. On the other hand, some regions of the NH basins, including the ENP, and NA basins show little linkage with simulated lower TC frequencies and higher low-level vorticity values (Supplementary Fig. S6C). Seasonal variations in the mid-level vertical velocity (Supplementary Fig. S6F) correspond with changes in relative humidity; the sustained convection that results from the greater mid-level moisture increases the mid-level upward motion. The lower values of the vertical wind shear between 850 and 200 hPa are important for the storms to maintain their strength, favoring the development of initial vortices to develop into storms and TCs. We further find that SH summer has higher shear in the MIS5e compared to the MIS5d in most of the ocean basins, which would oppose the simulated higher TC activities there (Supplementary Fig. S6G). In the paleoclimate simulations, the mismatch in the anomaly patterns of shear and TC frequency in some NH basins suggests that vertical wind shear plays only a minor role in describing global changes in TC frequency. This indicates that the variations in the thermodynamical variables, such as RH and MPI, are mainly responsible for the associated changes in TC formation across most of the global ocean basins in the paleoclimates, thereby outweighing the effects of other dynamical variables.

In response to rising $CO_2$ levels, the reduced RH values (Supplementary Fig. S12B) cause a decrease in TC frequency in most of the global ocean basins (except NA and ENP). We similarly observed reduced mid-level vertical velocity and large-scale vorticity in the SH summer in the $2\times CO_2$ experiment compared to the PD case (Supplementary Fig. S12C, F). The fluctuations of these conditions, however, exhibit basin asymmetry in the NH basins. Therefore, the low-level vorticity/mid-level vertical velocity variables cannot explain the TC formation differences in both hemispheres. In the $2\times CO_2$ experiment, the vertical wind shear variations exhibit larger values in SH summer

and asymmetric values in the NH ocean basins (Supplementary Fig. S12G). Hence, the underlying environmental conditions for a future warmer climate also suggest that moisture (RH) is a key factor influencing changes in the projected global TC frequencies.

## Changes in TC intensity
The diagnosed changes in the environmental conditions might have an impact on the TC intensity as well. Here we have further examined the changes in the lifetime maximum 10 m wind speeds and the mean sea level pressure of the storms in past, present, and future climate simulations (Supplementary Fig. S13). Although the model generally underestimates TC intensity, the model can capture the TC wind-pressure relationship in all the simulations and up to category-4 storms (Saffir-Simpson's scale) of the present climate and category-5 storms of future climate (Fig. 5 and Supplementary Fig. S13). From the MIS5d TC wind-pressure relationship, we can observe an increased frequency of intense storms compared to the MIS5e simulation. There is a clear intensification of storms across the globe in $2\times CO_2$ as compared to the PD climate.

The frequency of storms stratified by intensity category shows that there are less intense storms in the NH during the MIS5e, whereas more intense storms in the SH compared to the MIS5d. (Fig. 5). In contrast to the orbital forcing simulation, which has hemispheric asymmetry in the frequency of the intense TCs, the greenhouse warming simulation shows a higher number of intense storms in both hemispheres. Furthermore, there is a reduced frequency (−20 to −470% across the category) of NH intense TC categories (Category-2 to Category-4 storms) during the MIS5e and an increased frequency (+18 to +90%) in the SH during the MIS5e (Category-1 to Category-4 storms) compared to the MIS5d. In the $2\times CO_2$ simulation, we observe a reduction in the weaker category storms (i.e., tropical storms (TS) to Category-2 storms) and an increase in the frequency of the stronger storms (Category-3 to Category-5) in both the hemispheres as compared to PD condition. As there is an overall reduction in the global TC frequency for future warmer climate conditions compared to the past warmer climates (Supplementary Fig. S10), we observe a more pronounced reduction in the weaker storms in future climates as compared to the paleoclimate time-slice experiments. Therefore, in addition to the changes in the TC frequency, we also observe a hemispherically asymmetric response of TC intensities for the paleoclimate simulation and a symmetric response in a future warmer climate.

To further understand what controls the TC intensity (maximum of 10 m wind speeds), we examined one thermodynamical variable, the maximum potential intensity, which measures the convective instability of the atmosphere and gives the measure of maximum potential wind speeds a storm can attain under the background condition of the atmospheric temperature and moisture[1]. The seasonal differences in the maximum potential intensity, in general, agree well with TC intensity change (Supplementary Figs. S6E and S12E). In paleoclimate simulations, the seasonally varying radiation influences both atmospheric temperature and SSTs. The atmospheric temperatures respond faster than the underlying SSTs (Supplementary Fig. S14) to the increases in radiation, leading to reduced maximum potential intensity (Supplementary Fig. S15) in the NH summer of MIS5e as the upper troposphere is warmer while the surface layer is still not that heated yet. The NH autumn (October and November), by which time surface waters have several months to adjust, develops a warmer surface layer than the atmosphere above, with small differences leading to an increase in the maximum potential intensity (Supplementary Figs. S14 and S15). Therefore, maximum potential intensity values in the NH summer are greater in the MIS5d simulation than those in the MIS5e simulation, whereas maximum potential intensity in the SH summer is higher in the MIS5e simulation than in the MIS5d. These variations in the maximum potential intensity agree with earlier studies on paleoclimate that showed changes in TC intensity during the Holocene or Mid-Holocene periods[15,16]. In contrast, the

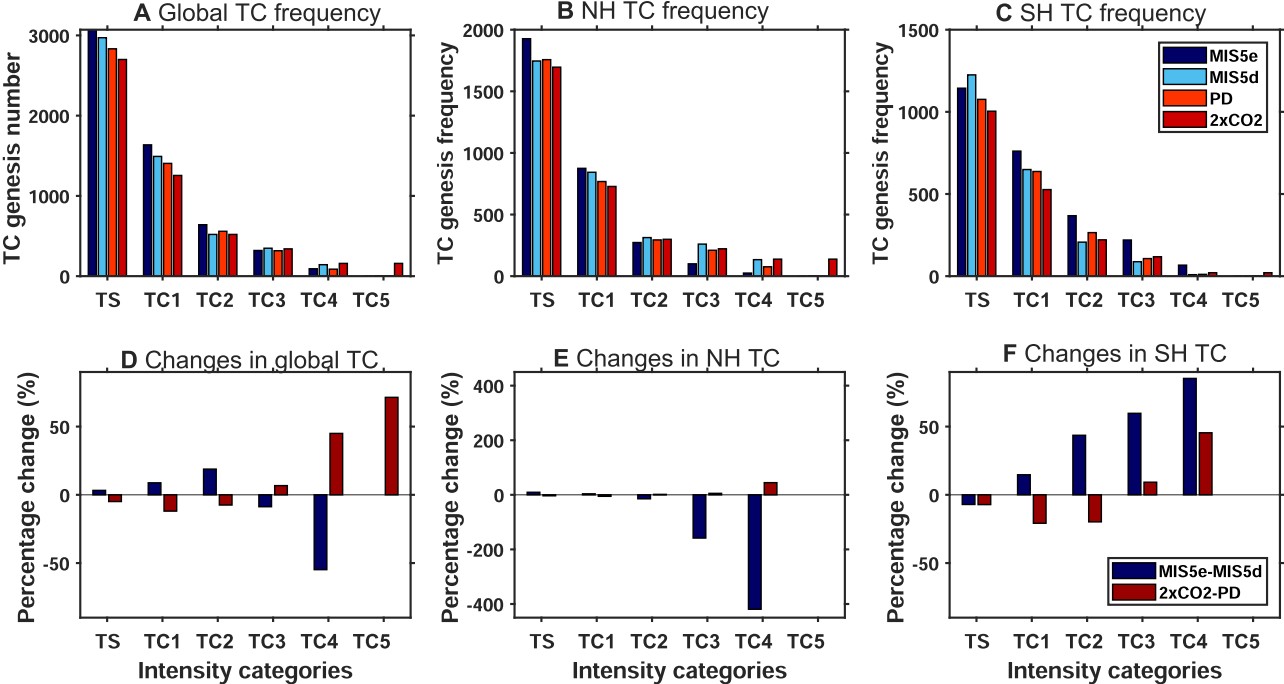

**Fig. 5 | The number of TC genesis and percent changes of TC genesis across different storm categories in the global, NH, and SH domains. A–C** TC genesis frequency counts and **D–F** percentage changes of TC frequency for storm categories from tropical storm (TS) to category 5 (TC5). The blue boxes in (**D–F**) represent the percent change due to orbital forcing (MIS5e minus MIS5d), and the red boxes represent the change due to greenhouse gas forcing ($2\times CO_2$ minus PD).

changes in maximum potential intensity for future climate simulation ($2\times CO_2$) follow the sign of SSTs throughout the year (Supplementary Figs. S14 and S15). Given these changes in the maximum potential intensity caused by orbital and $CO_2$ forcings, the NH summer in MIS5e cannot serve as an analog to the $2\times CO_2$ simulation because the environmental response at lower and upper levels varies significantly to different forcings.

## Discussion

Tropical cyclone (TC) intensity is predicted to increase owing to greenhouse warming by both theory and numerical simulations. However, it is not clear how changes in TC formation relate to different background climatological patterns. Therefore, conducting high-resolution model experiments that explicitly resolve TCs with varying climate conditions can enhance our understanding of the relationship between climate and TC activity. In the current study, we highlight the changes in TC genesis and intensity due to orbital and greenhouse gas forcings by explicitly simulating TCs under different background conditions using a mesoscale resolving coupled Earth system model. In response to orbital forcing, changes in TC frequency and intensity show a pronounced hemispheric asymmetry (Figs. 4 and 5). Both frequency and intensity increase in the SH summer of MIS5e, whereas the NH shows decreases in both characteristics except that frequency increases in the NI and ENP. In contrast, greenhouse warming leads to suppression in frequency and an increase in intensity in both hemispheres (Figs. 4 and 5).

An empirically derived GPI analysis captures most of the changes (except SI basin) in simulated TC frequency as obtained from two different tracking schemes qualitatively well under different background climate states (i.e., paleoclimate, present, and future) (Figs. 3 and 4, Supplementary Fig. S11). This supports the notion that changes in the model's large-scale climate variables can explain most of the associated changes in the TC activity. The seasonal composites of the large-scale climate variables show that in both the paleoclimate and future warmer climates, the moisture-related variables are the most important variables for the associated changes in the TC formations (Supplementary Figs. S6 and S12). Similarly, the GPI component analysis also shows that moisture-related variables (moist-entropy deficit and RH) explain the changes associated with TC frequency in paleoclimates and future climates (Table 2).

The earlier studies note that either increased saturation deficit/ entropy deficit of the atmosphere[30,31,41] or reduced upward mass flux[27] influence the TC frequency statistics for future warmer climates. Our study agrees with earlier studies[29,42], which note that both the mechanisms act simultaneously (i.e., increased moisture entropy deficit/reduced mid-level moisture reduces the upward mass flux) in explaining the response of TC frequency to climate change. In further examining the NH summer of the MIS5e and $2\times CO_2$ simulations, we observe a striking difference between the two simulations in terms of the movement of the ITCZ. In response to $CO_2$ forcing, the ITCZ shifts equatorward[11], whereas it shifts poleward in response to MIS5e orbital forcing (Supplementary Fig. S7). This poleward shift of the ITCZ during the peak TC season across the ENP and NI basins may lead to increased TC frequencies in the MIS5e as compared to the $2\times CO_2$ simulation (Fig. 4A). An earlier idealized study[43] noted an increase in the global TC frequency with warming due to poleward shifts of the ITCZ latitude which is larger than the reduction in TC frequency caused by the warming effect alone. The variations in the ITCZ position and the associated changes in large-scale thermodynamic variables during the NH summer of the MIS5e and $2\times CO_2$ simulations explain the differences in the simulated TC genesis activity. Seasonal temperature fluctuations at the surface and in the troposphere are related to variations in the theoretical intensity (maximum potential intensity) of paleoclimate TCs (Supplementary Figs. S14 and S15). Whereas for future climate conditions, the changes in SST can explain the variations in TC intensity. These results imply that the controlling factors and mechanisms for the changes in TC intensity are fundamentally different from paleoclimate to future climates.

Although the TC frequency response obtained from two fundamentally different tracking schemes is similar under various background

climate conditions, there is an underdetection of the storms in the main development region of the NA basin. The prevalence of SST biases in the Atlantic basin could have an impact on large-scale disturbances like the African Easterly waves and associated convective processes[6]. In summary, we infer that variations in tropospheric temperature and moisture are crucial attributes of the corresponding variations in TC activity in both paleoclimate and future climates. The limitation of this study is that it only used a single model. Future research using multiple models with equivalent forcings will better support the confidence in the conclusions from the current study.

## Methods
### Model description
In the current study, we employ a high-resolution CESM1.2.2[6] to perform fully coupled high-resolution global Earth system model simulations for paleoclimate, present and future climates. The atmospheric component of the model is CAM5 with a spectral dynamical core with a horizontal resolution of 0.25° and 30 vertical layers[44]. The ocean component is POP2 which has a horizontal resolution of 0.1° (decreasing from the equator (11 km) to poles (2.5 km)) and 62 depth layers[45]. The land model is the Community Land Model version 4[46], and the sea-ice component is the Community Ice Code version 4[47]. The time variations of the carbon-nitrogen cycle are switched off in the current model simulations. Chu et al.[6] shows that the TC climatology in the PD simulation is quite realistic, with certain regional and seasonal differences in TC genesis frequencies (Supplementary Fig. S5 of Chu et al.[6]) when compared with the observations (IBTrACS). They also show that the higher resolution reduces the mean SST bias (a common feature in some of the global climate models). The model can capture the localized air-sea interaction processes, mesoscale eddies, TC cold wakes, and convective features along the mountainous regions[6,19]. The details of the greenhouse gases, vegetation, and orbital years of the model simulations are shown in Table 1. The PD simulation was initialized from a quasi-equilibrium state of the Small et al.[19] and simulated for 140 years, the 2×CO$_2$ simulation was carried from the year 71 of the PD simulation with a doubling of atmospheric CO$_2$ concentration and integrated for 100 years. Similarly, the two paleoclimate simulations (MIS5e, MIS5d), which were branched off from year 71 of the PD simulation, are integrated for 100 years. In the current study, we carry out the analysis for a period of the last 60 years to check the robustness of the results. The main difference between the two paleoclimate simulations is the precession configuration. In the MIS5e, the Earth receives more solar insolation during the NH summer season and less during the SH summer season, and vice versa during the MIS5d period.

### Traditional TC detection and tracking scheme
Here we employ two different tracking schemes. The first is a traditional tracking scheme that uses certain circulation-based thresholds to identify and track initial vortices resembling TCs. This TC tracking scheme[6] is based on the idea of Bacmeister et al.[48], except that they use the 3-h interval data with a 200 km search radius for TC tracking and a 50 km radius for a 10 m wind speed threshold. The steps involved are as follows:
(i)   First, we identify the low-pressure systems using a threshold of surface pressure anomalies lower than −3 hPa;
(ii)  The 10 m wind speeds of the systems should exceed 10 m s$^{-1}$ within a radius of 100 km from the low-pressure minima location otherwise, those lows are discarded;
(iii) These detected lows at 6-h intervals are tracked forward in time based on the nearest location in a similar instance of time, and the tracking is continued if there is a low within a 400 km radius.
(iv)  A further check is involved to remove the duplicates of storm tracks and keep the tracks that have 17 m s$^{-1}$ magnitude of 10 m wind speeds during their lifetime and have a maximum duration of 2 days.

(v)   To obtain the global TC counts close to observations, we impose a further threshold of 0.00145 m s$^{-1}$ for the 850 hPa vorticity threshold.

### Phenomenon-based Okubo-Weiss-Zeta parameter (OWZP) tracking scheme
For the second scheme, rather than detecting TC-like vortices, a phenomenon-based tracking scheme uses the thresholds of the large-scale climate variables to identify the recirculating regions with stronger vorticity and weaker deformation that have the potential for TC formations. The OWZP scheme[49] uses the thresholds of large-scale climate variables such as relative humidity, specific humidity, vertical wind shear, and OWZP, which is equivalent to vorticity but involves stretching and shearing deformations. This TC detection and tracking scheme shows better performance for different TC characteristics as assessed using the ERA-interim and ERA5 data[49,50] and involves two steps:
(i)  In the first step, the "detection stage" involves identifying the grid points that satisfy the initial thresholds (see Supplementary Table S1 of the supporting information) and merging the nearby grid points into clumps;
(ii) In the next step, the "tracking stage," the detected clumps are tracked forward in time until there exists no circulation by using the storm position determined by the 700 hPa steering wind. These clump positions along each of the detected tracks are further checked using the core thresholds of Supplementary Table S1 and given True (thresholds are satisfied) and False (not satisfied). A particular track is considered a TC track if the core thresholds are satisfied for 48 h consecutively. In 12-h data, there should be five consecutive True clumps along the track and the fifth True position is considered as the storm genesis location[50].

### TC genesis potential index (GPI)
As TCs form under certain favorable environmental conditions, earlier studies developed certain empirical indices to identify the TC genesis locations. Emanuel and Nolan[51], Emanuel et al.[30], and Emanuel[31] developed GPIs with a set of environmental variables. These variables include (1) elevated values of large-scale absolute vorticity, which are required for the development of a strong localized TC-like vortex, and (2) higher mid-tropospheric moisture content, which is required for persistent convection through inflow from the moist boundary layer favoring deep convection. The mid-level moisture availability is measured through a magnitude of moist-entropy deficit estimated between the boundary layer and mid-troposphere and that changes with climate change even when the relative humidity does not change, (3) lower vertical wind shear which is necessary to maintain sustained vortex required for TC formation, (4) potential intensity which is the maximum attainable wind speed based on the available background thermodynamic environment of the atmospheric column (convective instability) involving the temperatures at the surface and outflow levels. Higher values of potential intensity indicate a more convectively unstable environment favoring the TC formation. Korty et al.[15] further developed this index with more sophisticated statistical analysis using similar environmental variables.

$$GPI(\text{Korty et al.}, 2012) = \left[\left(|\eta|, 4\times10^{-5}\right)\right]^3 \qquad (1)$$
$$[\max(V_{pot} - 35, 0)^2 \chi^{-\frac{4}{3}} [25 + V_{shear}]^{-4}$$

Term1 Term2 Term3 Term4

$$\chi = \frac{s_b - s_m}{s_o^* - s_b} \qquad (2)$$

Earlier GPIs are based on the SSTs, absolute vorticity, relative humidity, and vertical wind shear variables. As the SSTs are not

invariant with the background of changing climate conditions, this index cannot be employed in studies that focus on other climates[52]. The inclusion of 500 hPa vertical velocity can capture the future changes in the TC genesis activity[37].

$$GPI(\text{Murakami \& Wang, 2010}) = \left|10^5\eta\right|^{\frac{3}{2}}\left(\frac{RH}{50}\right)^3\left(\frac{V_{pot}}{70}\right)^3(1+0.1V_{shear})^{-2}\left(\frac{-\omega+0.1}{0.1}\right)$$

(3)

Term1 Term2 Term3 Term4 Term5

Here, $\chi$ is the moist-entropy deficit, $s_b$ is the boundary layer entropy, $s_m$ is the mid-tropospheric (600 hPa) entropy, $s_o^*$ is the saturation entropy at the sea surface, $\eta$ is the 850 hPa absolute vorticity, Vpot is the potential intensity, Vshear is the vertical wind shear between 850 and 200 hPa pressure levels, RH is the 700 hPa relative humidity, $\omega$ is the 500 hPa vertical velocity.

The above-discussed GPIs are evaluated in paleoclimate, present, and future climate studies[15,16,31,32,34–36]. These studies note that the modified GPIs with the inclusion of moist-entropy deficit/vertical velocity can capture the TC climatology and seasonal cycle in different climate conditions. A recent study by Camargo et al.[36] using a high-resolution 0.25° model shows that TC climatology estimated using GPI does not have a strong relationship with the model's explicitly simulated TC climatology. Most of the paleoclimate simulations are based on the changes in the GPI rather than TCs themselves, therefore in the current study, the comparison of the explicitly simulated TC climatology with the changes in the GPI can enhance our confidence in the paleoclimate TC variations and their associated atmospheric mechanisms.

### GPI component analysis

To decipher the relative role of each GPI term, we adopt a variational GPI analysis technique[53]. The equation below represents the changes in the TC genesis potential estimated using a GPI is equal to the sum of changes in each term of the GPI index by assuming that the non-linear (correlation) terms are negligible. Here the first term is the change in the variable between two simulations multiplied by the mean values of other terms of the GPI index. The summation of each of the terms on the right-hand side equals the overall changes in the GPI, that is, positive values correspond to the increasing GPI, and negative values reduce the GPI. In the current study, we conducted the GPI component analysis using both the GPIs.

$$\begin{aligned}\delta GPI(\text{Korty et al., 2012}) = &\ \delta Term1 \times (\underline{Term2 \times Term3 \times Term4})\\ &+ \delta Term2 \times (\underline{Term1 \times Term3 \times Term4})\\ &+ \delta Term3 \times (\underline{Term1 \times Term2 \times Term4})\\ &+ \delta Term4 \times (\underline{Term1 \times Term2 \times Term3})\end{aligned}$$

(4)

$$\begin{aligned}\delta GPI&(\text{Murakami \& Wang, 2010})\\ = &\ \delta Term1 \times (\underline{Term2 \times Term3 \times Term4 \times Term5})\\ &+ \delta Term2 \times (\underline{Term1 \times Term3 \times Term4 \times Term5})\\ &+ \delta Term3 \times (\underline{Term1 \times Term2 \times Term4 \times Term5})\\ &+ \delta Term4 \times (\underline{Term1 \times Term2 \times Term3 \times Term5})\\ &+ \delta Term5 \times (\underline{Term1 \times Term2 \times Term3 \times Term4})\end{aligned}$$

(5)

## Data availability

The observed tropical cyclone data is from the IBTrACS archive available at the following link (https://ncdc.noaa.gov/ibtracs/index.php?name=ib-v4-access) version 4. The HadISST is obtained from the Hadley Centre, UK Meteorological Office (http://badc.nerc.ac.uk/data/hadisst). The CESM code is publicly accessible from the National Center for Atmospheric Research. The model simulations using CESM 1.2.2 in the different climates are available upon request to the corresponding authors. All the data required to produce the graphics in the main manuscript and the supplementary materials are available upon request.

## Code availability

Figures are generated using the NCAR Command Language [version 6.4.0 (software), 2017, Boulder, Colorado: UCAR/NCAR/CISL/VETS; https://doi.org/10.5065/D6WD3XH5] and MATLAB software. The codes used to generate the figures are available upon request.

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

## Acknowledgements

This research was supported by the Institute for Basic Science, South Korea, under IBS-R028-D1. The model simulations are done using the CESM1.2.2, where the code is publicly available from the National Center for Atmospheric Research. Figures were generated by the NCAR Command Language (Version 6.4.0) [Software] (2017), Boulder, Colorado: UCAR/NCAR/CISL/VETS; https://doi.org/10.5065/D6WD3XH5. The simulations were conducted on the IBS/ICCP supercomputer "Aleph", 1.43 petaflops high-performance Cray XC50-LC Skylake computing system with 18,720 processor cores, 9.59 PB storage, and 43 PB tape archive space. We also acknowledge the support of KREONET.

## Author contributions

R.P.H., J.E.C., and A.T. planned the study, performed data analysis, and prepared the figures. R.P.H., J.E.C., and A.T. drafted and wrote the main manuscript. S.S.L. conducted the CESM1.2.2 simulations with different forcings. K.J.E. provided editorial assistance in improving the manuscript. R.P.H., J.E.C., A.T., S.S.L., and K.J.E. reviewed and edited the manuscript.

## Competing interests
The authors declare no competing interests.
