## [Peer Review File · Nature Communications]

REVIEWER COMMENTS

Reviewer #1 (Remarks to the Author):

The authors tried to give an idea of future tropical cyclone activity by comparing past and future climate simulations. They used the MIS5e, MIS5d scenarios for the past climate and the PD, CO2 doubling scenarios for the future climate. By comparing the responses of tropical cyclone activity to past and future climate, they found that moisture plays a critical role in tropical cyclone activity. In general, the manuscript is well organised. However, I have several concerns about its publication.

1) I found that the authors have a deep knowledge of paleo-climate studies. Since the NCC is a letter, I understand that the authors have tried to shorten the length of the manuscript. However, this makes it difficult for readers who are not experts in palaeo-climate to follow the significance of their findings. Since the NCC is a letter to the non-expert in the field, I suggest that the authors add more explanation and information about palaeoclimate studies.

For example, in lines 53-59, I don't know what is the limitation of the previous studies of warm climate episodes. Please provide more information.

Also, what is the difference between the MIS5e and MIS5d simulations compared to the last interglacial, the Eocene and the Pliocene simulations? Please explain and highlight the difference.

2) According to the introduction, it seems to me that the authors have found striking evidence in the projection of future tropical cyclone activity. However, their main finding is that only moisture is the most important. In fact, this result can only be found with the result of PD, CO2 doubling experiments. I don't know why the MIS5e and MIS5d experiments are essential to the authors' argument. Although the authors insist that the different experiments can reduce the uncertainty, I think the authors still cannot avoid the uncertainty problem by using the single model, CESM. I think the importance of this study is not to reduce the uncertainty, but rather a possible importance of tropospheric humidity.

3) Lines 155-174: I can acknowledge that RH related variables are most similar to GPI. If the quantitative contribution of each factor is given in the table, it will be much easier to understand the results.

Reviewer #2 (Remarks to the Author):

The following is a review of the manuscript submitted by Raavi et al entitled Moisture control of tropical cyclones in high-resolution simulations of paleoclimate and future climate. The results are interesting and provides new insights on the paleoclimatology of tropical cyclones, which are timely and relevant particularly in the advent of discussions surrounding global environmental changes. I have few comments, clarifications, and suggestions to the author. I believe the manuscript has potential for publication after the following comments have been adequately answered:

L65-66 - the use of NH summer vis-a-vis boreal summer here is confusing

L75 - What is MIS5d. It is not yet explained. why are they called extreme orbital conditions?

L98 - consider putting this explanation in earlier parts to provided clarity in the discussion

L148 - is it possible to put a timeseries of GPI in Fig. 1 to show such reduction in response to MIS5e-d, PDO, and 2xCO₂? This is for ease of reading for readers that are universally adept to paleoclimate.

L158 - "TC genesis"

L161 - WNP typically covers until 100E while NI covers until 95E

L167-168 - vertical wind shear in the WNP; how to determine the contribution of one term in GPI? is there a way to determine significance?

L171-173 - same with comments above. Looking at Fig. 3A and Fig S3C, there is a meridional component of dGPI extending from south to north in the WNP, which is not captured in Fig 3D but is captured in Fig. 3E. Same can be be said in the Caribbean and Eastern Pacific where VWS seems to affect the dGPI.

Similarly, the filament of negative 850 hPa abs. vorticity in Fig 3B, resembles that in the eastern Pacific indicating that atmospheric component (VWS and Abs. Vort) also affect dGPI. how to determine the contribution of one term in GPI? is there a way to determine significance?

L176-181 - There is a need to characterize the term "most" here because it can be seen that VWS also has considerable changes. This leads to my next question - does the blue mean decrease in VWS? If yes, does it mean more favorable TC genesis environment? Both sides remain to ask how to determine their significant contribution.

L197-198: this is confusing because I thought that the GPI in Fig. 3 denotes decrease in TC genesis. A northward ITCZ means favorable TC genesis environment because the moisture associated with the ITCZ is found in the WNP.

L215 - this is related to my previous comment. The mention of Fig S9 without relating to Fig. 4A first is confusing to the reader.

L264 - where in the figure does it show the correlation? is it through eyeball only?

L271 - what does it mean by global ocean changes in relation to VWS and TC frequency?

L272 - the mismatch in the anomaly patterns vis-a-vis GPI in two simulations show that thermodynamic is more important than dynamic variable? In a paper by Sharmila and Walsh (2017; <https://doi.org/10.1175/JCLI-D-16-0900.1>), they mentioned that dynamic parameters are more important in the WNP and South Pacific.

L295 - did you combine the peak season for each basin then plot the NH? Each basin has different peak season which has to be considered.

I think there is a need to provide more explanation/analysis on the influence and contribution of each GPI terms to justify that thermodynamics is more important. This is particularly true for WNP since it has the most TCs worldwide.

L555 - explain these acronyms, are MIS5e and 5d the gray columns only? kindly explain in the figures.

L585 - what does the line plot mean? kindly explain it in the caption. there is no x-axis label in Fig. 4A-B.

Reply to Reviewer's comments:

“Moisture control of tropical cyclones in high-resolution simulations of paleoclimate and future climate” by Pavan Harika Raavi, Jung-Eun Chu, Axel Timmermann, Sun-Seon Lee, and Kevin J. E. Walsh

Response to Reviewer 1:

Thank you very much for your valuable and helpful comments. We have answered your comments carefully and have made corrections according to your comments. The main response to your comments is as follows:

General comments

The authors tried to give an idea of future tropical cyclone activity by comparing past and future climate simulations. They used the MIS5e, MIS5d scenarios for the past climate and the PD, CO2 doubling scenarios for the future climate. By comparing the responses of tropical cyclone activity to past and future climate, they found that moisture plays a critical role in tropical cyclone activity. In general, the manuscript is well organised. However, I have several concerns about its publication.

Major comments

1. I found that the authors have a deep knowledge of paleo-climate studies. Since the NCC is a letter, I understand that the authors have tried to shorten the length of the manuscript. However, this makes it difficult for readers who are not experts in palaeo-climate to follow the significance of their findings. Since the NCC is a letter to the non-expert in the field, I suggest that the authors add more explanation and information about palaeoclimate studies. For example, in lines 53-59, I don't know what is the limitation of the previous studies of warm climate episodes. Please provide more information.

Response:

Thank you for your suggestion. We agree with the reviewer's comment that the limitations of the paleoclimate studies were not adequately addressed. We have expanded into more details about those paleoclimate studies in the introduction of the revised manuscript (Lines: **55-86**).

To highlight the several sentences, most of the studies are based on low-resolution models and/or the idea that large-scale environments such as an empirical TC genesis potential index (GPI) can represent changes in TC activity. To obtain a detailed perspective on the relationship between large-scale climatic drivers and TC statistics, it is important to explicitly resolve TCs and their key mesoscale features in a climate model. To the best of our knowledge, there are no systematic paleo-timeslice studies that explicitly resolve TCs using a fully coupled high-resolution Earth system model. Especially, simulation for MIS5e (LIG) has not been conducted. In other words, TC statistics such as frequency and intensity under different background conditions have not been fully tested.

This study is the first systematic approach to understand controlling factor of TC characteristics for both orbital forcing (MIS5e vs MIS5d) and greenhouse gas forcing (2xCO₂ vs PD) using fully coupled high-resolution CESM simulations. In addition, to improve the robustness of our results,

we also included two tracking schemes, two GPI indices, and a relatively long TC tracking period (60 years) for each simulation.

Also, what is the difference between the MIS5e and MIS5d simulations compared to the last interglacial, the Eocene and the Pliocene simulations? Please explain and highlight the difference.

Response:

The last interglacial (LIG) was the most recent warm period that took place at around 125ka and is also referred to as Marine Isotope Stage (MIS) 5e. Contrary to the present-day (PD) orbital configuration, the MIS5e was characterized by Northern Hemisphere (NH) summer perihelion conditions and warmer (colder) NH (SH) summers. The MIS5d is the glacial sub-stage characterized by Southern Hemisphere (SH) summer perihelion conditions and warmer (colder) SH (NH) summers. Stronger seasonal anomalies are caused by variations in the orbital forcing, which varies in the incoming solar radiation. A comparison of MIS5e and MIS5d helps us to understand the role of the orbital forcing on TC characteristics.

During the early-Eocene the global mean annual surface temperatures were 13 + 2.6 degrees warmer than the late 20th century temperatures and atmospheric CO₂ was 1400 ppmv. The Mid-Pliocene is the most recent period with the atmospheric CO₂ concentration comparable to the present (400 ppmv), global mean annual surface temperatures are 1.8 to 3.6 degrees higher than the pre-industrial temperatures.

In the LIG (MIS5e-MIS5d), global mean annual surface temperatures are 0.8-1.3 degrees warmer than the pre-industrial average with CO₂ levels of around 280 ppmv. The LIG period has lower CO₂ levels than other warmer epochs like the Eocene and Pliocene; the elevated temperatures are caused by significant seasonal variations in the incoming solar radiation and climate rectifications. The LIG period is the most recent time frame with higher global mean surface temperatures that can serve as an analogue to the current interglacial period.

These details are included in the revised manuscript at lines **55-65 and 89-98**.

2. According to the introduction, it seems to me that the authors have found striking evidence in the projection of future tropical cyclone activity. However, their main finding is that only moisture is the most important. In fact, this result can only be found with the result of PD, CO₂ doubling experiments. I don't know why the MIS5e and MIS5d experiments are essential to the authors' argument. Although the authors insist that the different experiments can reduce the uncertainty, I think the authors still cannot avoid the uncertainty problem by using the single model, CESM. I think the importance of this study is not to reduce the uncertainty, but rather a possible importance of tropospheric humidity.

Response:

We totally understand the reviewer's concern on uncertainty problem, and we appreciate this comment. In the revised manuscript, we tried to emphasize our efforts to improve robustness of our results.

Currently, there is no consensus on changes in future TC frequency. This brought us an idea to determine whether there is a common link between the large-scale environments and changes in TC characteristics from the past to the future climates. As the reviewer stated, our result is simple that moisture related variable or thermodynamical conditions are the dominating factors influencing the TC frequency both in the past and future warmer climates and we were happy to find this key factor.

It is noted that moisture related variable or thermodynamical conditions are the dominating factors influencing the TC frequency both in the past and future warmer climates. We may infer from the current study that one of the key contributing factors for TC frequency fluctuations related to various model climates (Past, present, and future) is moist entropy deficit.

The selection of MIS5e and MIS5d was made because these two periods represent the most recent extreme orbital conditions with low and high precession indexes respectively and large eccentricity. A low (high) precession index corresponds to NH summer perihelion (aphelion) condition and associated intensified (weakened) NH summer solar radiation. This effect is visible only for high values of Earth's orbit eccentricity and the intervals from 125-115 ka represent one of these periods. Therefore, MIS5e and MIS5d serve as excellent test grounds.

One of the limitations of this study is that we rely on a single model and a single ensemble. However, this model is among the highest TC-resolving scale fully coupled model and shows an excellent performance in representing TC characteristics and their air-sea interaction (Chu et al. 2020). To compensate for this limitation, we used a relatively long TC tracking period (60 years) for each simulation compared to previous work which used the last 20 years only. In addition, to improve the robustness of our results, we also included two fundamentally different tracking schemes and two different GPI indices. We hope you'll find this response to be acceptable.

3. Lines 155-174: I can acknowledge that RH related variables are most similar to GPI. If the quantitative contribution of each factor is given in the table, it will be much easier to understand the results.

Response:

Thank you for your good suggestion. In the revised manuscript, we included Table 2 that describes percentage contribution of the different GPI terms to total GPI change (MIS5e – MIS5d; 2xCO₂ – PD). The table clearly shows that moist related variable is the largest contributor among others.

Response to Reviewer 2:

We wish to thank the reviewer for insightful comments and suggestions on our manuscript, which were very useful to revise the first version. The followings are point-by-point responses to the comments:

General comments

The following is a review of the manuscript submitted by Raavi et al entitled Moisture control of tropical cyclones in high-resolution simulations of paleoclimate and future climate. The results are interesting and provides new insights on the paleoclimatology of tropical cyclones, which are timely and relevant particularly in the advent of discussions surrounding global environmental changes. I have few comments, clarifications, and suggestions to the author. I believe the manuscript has potential for publication after the following comments have been adequately answered:

L65-66 - the use of NH summer vis-a-vis boreal summer here is confusing

Response:

Thank you for your suggestion. The boreal summer is replaced with NH summer and the austral summer is replaced with SH summer.

L75 - What is MIS5d. It is not yet explained. why are they called extreme orbital conditions?

Response:

Thank you for raising this point. Based on this question raised by two reviewers, we learn that the explanation about the selection of MIS5e and MIS5d are not well addressed.

The MIS5d is a glacial sub-stage characterized by SH summer perihelion conditions and warmer (colder) SH (NH) summers. The selection of MIS5e and MIS5d was made because these two periods represent the most recent extreme orbital conditions with low and high precession indexes respectively and large eccentricity. The precession index is $e \sin(\omega_s)$, where e is the eccentricity and ω_s measures how close the sun is to the earth at midsummer. A low (high) precession index corresponds to NH summer perihelion (aphelion) condition and associated intensified (weakened) boreal summer solar radiation. This effect is visible only for high values of Earth's orbit eccentricity and the intervals from 125-115 ka represent one of these periods. Therefore, MIS5e and MIS5d serve as excellent test grounds.

L98 - consider putting this explanation in earlier parts to provided clarity in the discussion

Response:

Thank you for your suggestion. The detailed explanation of MIS5e and MIS5d are now added in the introduction part of the revised draft at lines **89-98**.

L148 - is it possible to put a timeseries of GPI in Fig. 1 to show such reduction in response to MIS5e-d, PDO, and 2xCO2? This is for ease of reading for readers that are universally adept to paleoclimate.

Response:

Thank you for your comment. We do agree that adding the GPI time series will ease the readers. However, our experiments (MIS5e, MIS5d, PD, and 2xCO₂) are paleo and future timeslice simulations using a high-resolution CESM model with 0.25° in the atmosphere and 0.1° in the ocean, whereas the time series in Fig. 1 are from the long-term transient climate simulation using a low-resolution CESM with the horizontal resolution of 3.75°×3.75° (Timmermann et al. 2022). Therefore, the experiments are fundamentally different. To avoid confusion from two different simulations, we would like to keep Fig. 1 in its current form. We hope the reviewer understands our intention.

L158 - "TC genesis"

Response:

Thank you for your comment. It is corrected in the revised manuscript.

L161 - WNP typically covers until 100E while NI covers until 95E

Response:

Thank you for your comment. This is corrected in the revised manuscript for the respective figures.

L167-168 - vertical wind shear in the WNP; how to determine the contribution of one term in GPI? is there a way to determine significance?

Response:

Thank you for your comment. In the revised manuscript, we included Table 2 that describes percentage contribution of the different GPI terms to total GPI change (MIS5e – MIS5d; 2xCO₂ – PD) with significant test. The table clearly shows that moisture related variable is the largest contributor among others.

L171-173 - same with comments above. Looking at Fig. 3A and Fig S3C, there is a meridional component of dGPI extending from south to north in the WNP, which is not captured in Fig 3D but is captured in Fig. 3E. Same can be said in the Caribbean and Eastern Pacific where VWS seems to affect the dGPI.

Response:

Thank you for pointing this out. As the reviewer stated, VWS plays an important role in determining dGPI over the western North Pacific, eastern Pacific, and Atlantic Ocean, especially in response to the orbital forcing (i.e., MIS5e-MIS5d). Therefore, their role is not negligible on the regional scale. However, the contribution of VWS is limited to the off-equatorial zone at around 10°N, making its overall contribution over the entire NA and WNP less important than MED (Table 2). In addition, the contribution of VWS in response to greenhouse gas forcing is not statistically significant (Table 2).

Similarly, the filament of negative 850 hPa abs. vorticity in Fig 3B, resembles that in the eastern Pacific indicating that atmospheric component (VWS and Abs. Vort) also affect dGPI. how to determine the contribution of one term in GPI? is there a way to determine significance?

Response:

Our response to this question will be similar to the above answer. Likewise in vertical wind shear, the 850 hPa absolute vorticity seems to play an important role in the off-equatorial zone. But its contribution over the entire basin is less important and is not statistically significant in both forcings. As this study aims to determine whether there is a common link between the large-scale environments and changes in TC characteristics that works both in the past to the future conditions, we emphasized more on the moist-entropy deficit (MED). But as we mentioned, VWS and AVOR are also important in some areas.

The percent contribution in Table 2 is calculated by dividing the relative role of each GPI term (each term on the right-hand side of the GPI component analysis) by total changes in GPI (dGPI). Because the GPI has no units, we expressed the percentage contribution rather than the actual dGPI value. The statistical significance of each term at individual grid points was tested by calculating GPIs by replacing only one term from each simulation. For example, the significance of the MED contribution between MIS5e and MIS5d was calculated by comparing two GPIs: one with all terms in the MIS5e simulation and the other with all terms in the MIS5e simulation except for MED in the MIS5d simulation.

L176-181 - There is a need to characterize the term "most" here because it can be seen that VWS also has considerable changes. This leads to my next question - does the blue mean decrease in VWS? If yes, does it mean more favorable TC genesis environment? Both sides remain to ask how to determine their significant contribution.

Response:

We apologize for any confusion caused due to insufficient explanation. Colors in Fig. 3 from individual terms indicate their contribution to dGPI. For example, dGPI in Fig. 3A is the sum of Fig. 3B to Fig. 3E. Red colors in each term indicate favorable TC genesis environment and blue colors indicates unfavorable condition. For example, blue colors in vertical wind shear term (e.g.,

Fig. 3E) means decreases in Term4 (i.e., $(25 + V_{shear})^{-4}$) which is driven by increased vertical wind shear. Regarding the significant contribution, we believe the newly added Table 2 will help in understanding better.

L197-198: this is confusing because I thought that the GPI in Fig. 3 denotes decrease in TC genesis. A northward ITCZ means favorable TC genesis environment because the moisture associated with the ITCZ is found in the WNP.

Response:

You are right. In general, unfavorable conditions (decreases in TC generation) are produced in the NH summer due to changes in the thermodynamic environment in response to orbital forcing. However, a northward-shifted ITCZ partly offsets this effect, and we found that this influence is limited to the NI and ENP regions, not in the WNP basin (Fig. R1).

Fig. R1. Differences in (a) GPI and (b) precipitation between MIS5e and MIS5d during the TC peak seasons across different ocean basins (i.e., JAS in North Pacific and North Atlantic basins; JFM in Southern Hemisphere basins; OND in North Indian Ocean).

L215 - this is related to my previous comment. The mention of Fig S9 without relating to Fig. 4A first is confusing to the reader.

Response:

Thank you for your valuable comment. We tried to increase the readability in the figure transition between Fig. 4 and Fig. S9 in lines **245-257**.

L264 - where in the figure does it show the correlation? is it through eyeball only?

Response:

Thank you for pointing out this inadequate word “correlate”. The sentence is modified in lines **300-302** as below.

“Seasonal variations in the mid-level vertical velocity (Fig. S6F) correspond with changes in relative humidity;”

L271 - what does it mean by global ocean changes in relation to VWS and TC frequency?

Response:

The “global ocean changes” is now corrected to “global changes” in line **308**. Thank you.

L272 - the mismatch in the anomaly patterns vis-a-vis GPI in two simulations show that thermodynamic is more important than dynamic variable? In a paper by Sharmila and Walsh (2017; <https://doi.org/10.1175/JCLI-D-16-0900.1>), they mentioned that dynamic parameters are more important in the WNP and South Pacific.

Response:

Thank you for your comment. The dynamical factors play a role in the orbital forcing simulations. However, we looked for a consistent or common limiting factor that applies to the majority of the global basins both in past and future warmer climates, and we emphasize that thermodynamic factor is the major controlling factor.

L295 - did you combine the peak season for each basin then plot the NH? Each basin has different peak season which has to be considered. I think there is a need to provide more explanation/analysis on the influence and contribution of each GPI terms to justify that thermodynamics is more important. This is particularly true for WNP since it has the most TCs worldwide.

Response:

Thank you for your comments. We have considered different peak seasons for different ocean basins, including JAS for the NA, ENP, and WNP; OND for the NI; and JFM for the SI and SP basins. As we have many components from two forcings, we wanted to exhibit them on one map instead of showing all seasons in separate figures. In the revised manuscript, we have included Table 2 to quantitatively identify the governing factors associated with the TC frequency changes

in both orbital and greenhouse warming experiments. We also tested that changing one month before and after the peak season in the WNP does not change our conclusions.

L555 - explain these acronyms, are MIS5e and 5d the gray columns only? kindly explain in the figures.

Response:

Thank you for your comments. The gray bars indicate the timing for four timeslice simulations. We included this information in the Fig. 1 caption of the revised manuscript.

L585 - what does the line plot mean? kindly explain it in the caption. there is no x-axis label in Fig. 4A-B.

Response:

Thank you for pointing out this missing information. The blue line plot shows the zonal mean latitudinal distribution of TC frequency changes. We added this information to the figure caption and plotted the x-axis label in the updated revised manuscript.

REVIEWERS' COMMENTS

Reviewer #1 (Remarks to the Author):

I appreciate the authors' efforts to respond to my comments. All my comments are well addressed, but there is just one point where I feel a bit uncomfortable.

I think the importance of this study is to show that moisture is the most controlling factor for tropical cyclone frequency under global climate change, regardless of whether the origin of climate change is greenhouse gases or other external forcings. I think this is a very interesting finding and may provide clues to understanding future tropical cyclone activity. However, this is only true for the single model used in this study, which is the limitation of this study as the authors replied. In this respect, I don't agree that this study contributes to reducing uncertainty about future changes in tropical cyclone activity. In the revised version of the manuscript, the related argument is weakened, so I don't have a strong disagreement against the revised manuscript. Nevertheless, I would recommend that the authors avoid the statement about "uncertainty" in the introduction if they can. This is because the statement may mislead readers into expecting that their study reduces uncertainty problems. Another alternative is for the authors to make the limitations of their study clearer in the conclusion.

Reviewer #2 (Remarks to the Author):

This is a review of the manuscript submitted by Raavi et al entitled Moisture control of tropical cyclones in high-resolution simulations of paleoclimate and future climate.

I believe that the results are interesting and provides new insights on the paleoclimatology of tropical

cyclones. I believe that the authors have adequately replied to my comments in the previous revision. I have minor editorial comments to which I defer to the editors for their discretion on such comments. I would like to see bigger font/typeface size in ALL FIGURES (both main and supplementary) as it is quite challenging to squint through the manuscript. Where it is possible, use bigger font sizes as some readers prefer printed copy rather than electronic copy. Make sure that the color palette are also color blind-safe. The green in Fig. 5 is almost difficult to see. The contour in Fig. S1 is too thick compared to the rest of the image. After all of these minor editorial comments have been answered, I believe the manuscript can be considered for publication.

Response to Reviewers

“Moisture control of tropical cyclones in high-resolution simulations of paleoclimate and future climate” by Pavan Harika Raavi, Jung-Eun Chu, Axel Timmermann, Sun-Seon Lee, and Kevin J. E. Walsh

Dear Editor:

We appreciate the opportunity to address the reviewers’ constructive suggestions and revise our manuscript “Moisture control of tropical cyclones in high-resolution simulations of paleoclimate and future climate” once again. We appreciate the editor and the reviewers for their thorough reviews and useful comments that have been of great help in improving the manuscript. A response to the individual comments is provided below, with the reviewer’s comments in bold-faced text and our answers in the light-faced text. Finally, we thoroughly checked the editorial requests and revised our figures and texts carefully.

Yours sincerely,

Pavan Harika Raavi and Jung-Eun Chu

Reviewer #1 (Remarks to the Author):

I appreciate the authors' efforts to respond to my comments. All my comments are well addressed, but there is just one point where I feel a bit uncomfortable. I think the importance of this study is to show that moisture is the most controlling factor for tropical cyclone frequency under global climate change, regardless of whether the origin of climate change is greenhouse gases or other external forcings. I think this is a very interesting finding and may provide clues to understanding future tropical cyclone activity. However, this is only true for the single model used in this study, which is the limitation of this study as the authors replied. In this respect, I don't agree that this study contributes to reducing uncertainty about future changes in tropical cyclone activity. In the revised version of the manuscript, the related argument is weakened, so I don't have a strong disagreement against the revised manuscript. Nevertheless, I would recommend that the authors avoid the statement about "uncertainty" in the introduction if they can. This is because the statement may mislead readers into expecting that their study reduces uncertainty problems. Another alternative is for the authors to make the limitations of their study clearer in the conclusion.

Thank you very much for your valuable comments. In the revised manuscript, we included a statement regarding the limitation of the study which is based on single model usage (Line 416–418). In addition, to avoid overstating the uncertainty issue, we removed the statement about uncertainty reduction in the introduction of the revised manuscript. Thank you.

Reviewer #2 (Remarks to the Author):

This is a review of the manuscript submitted by Raavi et al entitled Moisture control of tropical cyclones in high-resolution simulations of paleoclimate and future climate. I believe that the results are interesting and provides new insights on the paleoclimatology of tropical cyclones. I believe that the authors have adequately replied to my comments in the previous revision. I have minor editorial comments to which I defer to the editors for their discretion on such comments. I would like to see bigger font/typeface size in ALL FIGURES (both main and supplementary) as it is quite challenging to squint through the manuscript. Where it is possible, use bigger font sizes as some readers prefer printed copy rather than electronic copy. Make sure that the color palette are also color blind-safe. The green in Fig. 5 is almost difficult to see. The contour in Fig. S1 is too thick compared to the rest of the image. After all of these minor editorial comments have been answered, I believe the manuscript can be considered for publication.

Thank you for your considerate comments. To enhance the readability, we carefully checked the font sizes, information, and blind-safe colors for all figures in the revised manuscript, including Fig. 5 and Fig. S1 as you suggested. Thank you.